# Effects of Weaning Age and Creep Feed Type on Growth Performance and Gut Maturation in Weaned Piglets

**DOI:** 10.3390/ani13111851

**Published:** 2023-06-01

**Authors:** Kimmie Kyed Lyderik, Xuwen Zhang, Christina Larsen, Niels Jørgen Kjeldsen, Marie Louise Madelung Pedersen, Mette Skou Hedemann, Andrew Richard Williams, Charlotte Amdi, Johannes Gulmann Madsen

**Affiliations:** 1Department of Veterinary and Animal Sciences, Faculty of Health and Medical Sciences, University of Copenhagen, Grønnegårdsvej 2, 1870 Frederiksberg, Denmark; kikl@sund.ku.dk (K.K.L.); xuwenmegan@gmail.com (X.Z.); christinalarsen@sund.ku.dk (C.L.); arw@sund.ku.dk (A.R.W.);; 2SEGES Innovation P/S, Axeltorv 3, 1609 Copenhagen, Denmark; kjeldsenniels68@gmail.com (N.J.K.); mlp@seges.dk (M.L.M.P.); 3Department of Animal and Veterinary Sciences, Faculty of Technical Sciences, Aarhus University, Blichers Alle 20, 8830 Tjele, Denmark; mette.hedemann@anivet.au.dk

**Keywords:** piglets, weaning, enzyme activity, absorption, growth performance, liquid feed

## Abstract

**Simple Summary:**

The transition from suckling to weaning imposes challenges for piglets and often results in unwanted post-weaning growth check. This is a result of poor feed intake, which is likely a consequence of piglets’ lack of adaption to a dry vegetable-based diet. To accommodate a smoother transition, the objective of this study was to investigate the effects of four- vs. five-week weaning ages and dry vs. liquid feeds, to resemble the texture of sow’s milk, on growth performance, the activity of certain disaccharidases and the gene expression of nutrient transporters in the small intestines of weaned pigs. Later weaning and liquid feed resulted in heavier pigs during the later weaning phase. In conclusion, a clear increase in the growth performance of liquid-fed pigs was not associated with changes in the investigated disaccharidase activity or gene expression of the analyzed nutrient transporters.

**Abstract:**

The objective was to study the effects of weaning in week 5 (W5) vs. week 4 (W4), as well as liquid (LF) vs. dry feed (DF), on growth performance, disaccharidase activity and nutrient transporter expression after weaning. The experiment included 12,923 pigs fed LF or DF in the pre-weaning period and a subpopulation of 15 pigs from each group, W4DF, W4LF, W5DF and W5LF, which were weighed and euthanized five days after weaning. The proximal part of the small intestine was analyzed for maltase, lactase and sucrase activity and the expression of *SGLT-1*, *GLUT-2* and *PepT-1*. Pigs fed LF displayed less maltase activity (2100 vs. 2729 U/mg protein, *p* < 0.05) but an increased expression of *SGLT-1* (∆Ct: 5.22 vs. 6.21, *p* = 0.01). Pigs weaned in W5 were heavier than those weaned in W4 (9.35 vs. 7.11 kg BW, *p* ≤ 0.05), and pigs fed LF were heavier than those fed DF (8.55 vs. 7.91 kg BW, *p* ≤ 0.05) five days after weaning in the subpopulation. LF pigs (21.8 kg) were heavier than DF pigs (20.6 kg) (SE 0.108, *p* < 0.0001), and W4 pigs (21.0 kg) were lighter than W5 pigs (21.5 kg) (SE 0.108, *p* = 0.01) at nine weeks. LF increased weight gain in the early post-weaning period and at nine weeks, although this was apparently not explained by accelerated gut maturation.

## 1. Introduction

The period of transition from suckling on sow’s milk to feeding on a dry vegetable-based diet in the early weaning phase is critical for the piglet’s growth performance, health status and intestinal development [1,2]. Because commercially produced piglets are weaned relatively early compared with those observed under semi-natural conditions [3], and because the developmental stage of the gut, as a physiological feature, has been shown to depend more on age than on body weight [4,5,6,7], questions have been raised about pigs’ abilities to digest and utilize the more complex feed provided after weaning. More specifically, the activity of digestible enzymes, namely maltase, sucrase and lactase, have proved to be of great interest as indicators of a pig’s ability to adapt to different carbohydrate sources in the diet [8,9]. Furthermore, a pig’s ability to abruptly shift from milk with a dry matter (DM) content of ~18–20% [10,11] to a dry diet with an ~85% DM has also been questioned. The latter has been associated with the frequently observed growth check caused by feed refusal during the first few days after weaning [12]. Feed refusal leads to the deterioration of villi in the small intestine (SI), which further impairs digestible capacity and prolongs growth check [13]. Thus, it is imperative that piglets either adapt to other sources of nutrients besides sow’s milk or that the gut reaches a more mature state prior to weaning.

Several studies have investigated how the early provision of creep feed during the suckling period affects growth performance in the post-weaning period. Van Der Meulen et al. [14] found that creep feeding familiarized pigs with solid feed and produced more eaters. Additionally, it was previously observed that the physical characteristics of feeding treatments, such as play feeders [15] and larger sizes of pelleted feed [16], positively affected feed intake and eased the weaning transition. The provision of liquid creep has scarcely been investigated; Byrgesen et al. [17] showed greater feed disappearance when piglets were fed liquid feed compared with dry creep feed. However, this observation was not followed by an increase in disaccharidase activity around weaning nor an increase in growth performance during the post-weaning period. While some research has been carried out to investigate the impact of weaning age on growth performance [18], no studies have reported on comparisons of four- and five-week weaning ages and the effect of the feeding strategy.

Therefore, the present study aimed to explore the effects of a one-week increase in weaning age, the impact of liquid form feeding treatment compared to dry feed and the interaction between weaning age and feed treatment on growth performance and intestinal development after weaning. It was hypothesized that providing liquid creep feed would increase pre-weaning feed intake, thereby accelerating maturational changes in the small intestine and increasing the digestive and absorptive capacity of pigs. Additionally, it was hypothesized that an increased weaning age would promote maturational changes in the small intestine through both increased feed intake and age-related development.

## 2. Materials and Methods

### 2.1. Ethical Approval

All procedures involving animals were conducted in accordance with the guidelines of the Danish Ministry of Justice with respect to animal experimentation and the care of animals under study (The Danish Ministry of Justice, 1995) and were approved under the following AEIRB approval number: 2021-10-PNH-018A.

### 2.2. Animals and Exprimental Design

To elucidate the effects of weaning age and liquid versus dry creep feed on the pre- and post-weaning performance of commercially reared pigs, a large-scale trial was conducted in a functioning Danish sow herd producing (Landrace × Yorkshire) × Duroc crossbred piglets. The large-scale trial included 12,923 piglets at weaning age and was conducted from March 2020 to February 2021. The experimental period lasted from post-partum day three until the pigs reached nine weeks of age.

To determine whether age and creep feeding promoted maturational changes in the digestive system, two subpopulations of pigs were selected and euthanized for organ and tissue sampling according to weaning age and the type of creep feed provided during their stay in the farrowing unit. Sixty pre-weaning piglets were sacrificed three days prior to being weaned, and the results of the pre-weaning subpopulation are presented in Lyderik et al. [19]. The subpopulation described in the present study consisted of 60 post-weaning pigs euthanized in the immediate post-weaning period. The sampled pigs were born from 54 different sows ranging from 2nd to 8th parity.

The present study included 4 treatment groups of 15 weaned pigs in a 2 × 2 factorial design. Each treatment consisted of a combination of a given weaning age (WA) and creep feeding strategy (FS), where pigs were weaned in the 4th or 5th postnatal week (W4 or W5) and received either dry or liquid creep feed (DF or LF). Therefore, the 4 treatment groups were W4DF (n = 15), W5DF (n = 15), W4LF (n = 15) and W5LF (n = 15). Extending the lactation period by one week and providing liquid creep feed are realistic and implementable interventions which might ease the weaning transition by allowing the gut to ontogenetically mature through a potential increase in piglet feed intake. Another important consideration was to include cost-effective treatments so as to ensure greater implementation with potential to reduce antimicrobial use.

The mean weaning age was 24.5 ± 0.8 days for pigs in the W4 group and 31.6 ± 0.9 days for pigs in the W5 group. The pigs were euthanized 5 days post-weaning; thus, the W4 pigs had a mean age of 29.5 ± 0.8 days at euthanasia, whereas the W5 pigs were 36.6 ± 0.9 days of age. All pigs in the large-scale trial, including those in the sampled post-weaning subpopulation, were weighed at post-partum day three and again at weaning, and the feed allowance per double pen was monitored daily.

### 2.3. Housing and Management

The commercial herd had an average production size of 1170 year-sows and weaned approximately 800 pigs per week. In the farrowing unit, sows and litters were kept in traditional farrowing pens with crates and partially slatted floors.

To stimulate the ingestion of creep feed through hunger, a sow nursed one additional piglet more than her number of functional teats. Therefore, the litter size was determined by the number of functional teats on day 3, when the litters were locked. The litter size on day 3 varied from 9–17 piglets per sow. Shortly after birth, all piglets received Vetrimoxin treatment (Ceva Aninal Health, Libourne, France), and two days post-partum, the piglets were treated with Dozuril (Dopharma Research, Raamsdonksveer, The Nederlands), and male piglets were castrated.

In the weaner unit, pigs from the same treatment group were reared in connected double pens and littermates were kept together; thus, the stocking density varied from 31 to 49 pigs per pen. One-third of the pen floor was solid and served as a covered creep area. The rest of the floor was slatted concrete.

### 2.4. Diet and Feeding Regime

To ensure that the piglets’ nutrient and energy requirements were met, all litters had access to milk supplement (Danmilk™ Supreme 1.0, Agilia™, Videbæk, Danmark) during the first 8 postnatal days (MS1). Litters receiving the LF diet had access to a second milk supplement (MS2; Danmilk™ Gain Agilia™, Videbæk, Denmark) from day 9 to 17. The compositions and nutrient contents of MS1 and MS2 can be seen in Table 1.

Regarding pre-weaning creep feeding, from post-partum day 9 to the time of weaning, the W4DF and W5DF piglets had access to a dry zinc-free weaner diet as a creep feed (weaner diet 1). Piglets in groups W4LF and W5LF received MS2 from post-partum days 9 to 17, and from post-partum day 18 to weaning, they received weaner diet 1 but as a liquid feed. Here, the dry feed was mixed with water so that the DM constituted 20%. The diet compositions can be seen in Table 2.

Computerized Mini Wet Feed Systems (BoPil A/S, Sønderborg, Denmark) were installed in the farrowing pens where sows reared LF litters. Feeding troughs were placed above the slatted floor and in the separating wall between two mirrored and connected farrowing pens. Thus, two litters shared one feeder. To ensure continuous access to creep feed, the troughs were equipped with sensors registering when they were empty. When this occurred, 20% of the daily ration would be allocated. Both MS2 and LF were allocated in this manner. The computer registrations of the allocated feed amounts were used to estimate the average feed allowance. In the DF pens, piglets received their feed through separate computerized feeding pipes, and the allocated amounts were therefore registered separately. For the DF piglets, the dry feed was served in the same trough as MS1, and both were provided 5–6 times per day, ensuring constant availability.

As for the post-weaning diet and feeding regime, all weaned pigs continued to receive the same feed that they had received as creep feed during the first eight post-weaning days. From post-weaning days 9 to 14, the DF pigs were gradually transferred to LF feed. All pigs then received the same liquid weaner diet 1 until they reached 18 kg in body weight (BW). From 18kg and onwards, all pigs received a second weaner diet (weaner diet 2). All weaner diets were formulated based on the national nutritional standards set by SEGES Innovation P/S [20], and their compositions can be seen in Table 2. In the weaner unit, connected double pens shared a feeder, and therefore, the estimated feed allocation is based on the registrations per double pen.

### 2.5. Organ and Tissue Sampling

Pigs were euthanized 5 days post-weaning on the same weekday for four consecutive weeks. For each of the first three weeks, 16 pigs were sacrificed, and for the last week, 12 pigs were sacrificed in one day. Thus, a total of 60 pigs were sacrificed.

Sex and litter origin were included as selection criteria. Thirty males and thirty females were included, having been chosen evenly across treatment groups. The aim was to include 1 random pig from 60 different litters. Unfortunately, this was not possible, and therefore 2 pigs were selected from 6 randomly chosen litters of the 54 litters available. Otherwise, pigs were randomly chosen. Stocking density was not included in the selection criteria, as the present study aimed to reflect general on-farm conditions in which stocking density varies greatly from batch to batch.

Before euthanasia, the BW was registered. Pigs were euthanized using a bolt pistol and were exsanguinated before the removal of the entire gastrointestinal tract. The stomach, SI and colon were emptied and weighed, and the length of the SI was measured.

Tissue samples of the proximal SI were collected from each of the post-weaning pigs for a later analysis of enzyme activity and gene expression. This sampling site was chosen based on earlier findings by Amdi et al. [8,9], where the inclusion of complex carbohydrates in a pre-weaning milk replacer affected the proximal part of the SI but not the medial or distal part. The proximal sampling site was located by dividing the entire SI into 3 sections of equal length. The section cut near the stomach was considered as the proximal part. This section was then folded to find the middle, and from this area, two 0.5 cm samples were collected.

Tissue samples intended for enzyme activity analysis were rinsed with water and immediately placed in cryotubes on dry ice and later stored at −80 °C, while samples for the analysis of gene expression were placed in cryotubes with RNAlater™ (Invitrogen, Thermo Fischer Scientific, Vilnius, Lithuania) and stored at −20 °C.

### 2.6. Analysis of Enzyme Activity

The activities of maltase, sucrase and lactase were determined for each SI tissue sample homogenized in 1% Triton X-100. The tissue to Triton X-100 ratio was 0.20 mg tissue per 1 mL solution. Using the method described by Dahlqvist (1968) [21] to determine disaccharidase activity, the concentration of generated glucose was measured via the absorbance at 340 nm using an EL × 808 microplate reader (Bio-tek, Winooski, VT, USA), and the generated quantity of glucose was determined based on a nicotinamide adenine dinucleotide phosphate (NADP) chain reaction:$$Glucose+ATP\;\overset{\mathrm{HK}}{\to }\;Glucose\;6\text{-}phosphate+ADP\;Glucose\;6\text{-}phosphate+NADP\;\overset{G\text{-}6\text{-}P\text{-}\mathrm{DH}}{\to }\;Gluconate\text{-}6\text{-}phosphate+NADPH+H^{+}$$

For each sample, enzyme activity measured in Units per liter (U/L) was calculated as:∆c/t = (∆E × a mL × d)/((t × b (L × mol^−1^ × cm^−1^) × c cm × e µL) × n)
= (∆E × 3.22 mL × d × 10^6^ µmol/mol)/((60 min × 6300 (L × mol^−1^ × cm^−1^) × 0.93 cm × 0.1 µL) × n)
= (∆E × 91.597 × d)/n

In this calculation, ∆c/t denotes the change in glucose concentration over the reaction time, and ∆E is the change in absorption measured per minute. The a signifies the total volume in a well, d is the dilution factor for the specific sample, and the extinction coefficient of NADPH (6300 M^−1^ cm^−1^) is denoted as b. The well depth is denoted as c, and e is the sample volume. The number of glucose molecules generated per molecule of hydrolyzed disaccharide is denoted as n. For maltose, n = 2, while n = 1 for sucrose and lactose. Before performing any statistical analysis of the results, the unit U/L was converted to U/mg tissue by multiplying ∆c/t with 0.2 mg tissue per mL Triton X-100 solution.

A single sample from a pig in the W5LF treatment group contained an insufficient amount of tissue to be analyzed for all three enzymes, and therefore, maltase activity was not determined for this sample. Thus, maltase activity was determined for 59 pigs, while sucrase and lactase were determined for all 60 pigs.

### 2.7. Analysis of Nutrient Transporter Gene Expression

Using relative quantification and real-time polymerase chain reaction (RT qPCR), the gene expression of sodium-glucose linked transporter 1 (*SGLT-1*), glucose transporter 2 (*GLUT-2*) and peptide transporter 1 (*PepT-1*) was analyzed.

RNA was extracted from the SI tissue samples using a lysis reagent and RNeasy mini kits (Qiagen, Hilden, Germany) following the accompanying protocol from the manufacturer. For each of the 60 samples, 500 ng of RNA was used to synthesize cDNA with a Reverse Transcription kit (Qiagen, Hilden, Germany). An Aria MX PCR system (Agilent Technologies, Santa Clara, CA, USA) was used to perform the qPCR analysis using the following cycling program: 2 min hot start at 95 °C, forty amplification cycles of 5 s at 95 °C and 20 s at 60 °C, and a melt cycle of 3 × 30 s at 95 °C, 65 °C and 95 °C, respectively. The primer sequences can be seen in Table 3.

The primer efficiency was evaluated through the construction of linear standard curves for each primer pair. Porcine Beta-2-Microglobulin (*B2M*) served as the reference gene due to its stable expression in the intestinal tissue samples, according to which the normalized relative genetic expression of nutrient transporter genes was calculated. The ∆Ct value of each gene was calculated for each of the original tissue samples and was chosen as a parameter because of its ability to be analyzed using linear modelling and regression [22]. When interpreting ∆Ct results, it is important to consider the inverted relationship between ∆Ct and the gene expression level, where a lower ∆Ct value signifies a higher level of gene expression.

Despite repeated attempts, three samples could not be analyzed for gene expression. Additionally, another tissue sample did not contain enough tissue to be analyzed for all three genes and was therefore not analyzed for PepT1 gene expression. Hence, 57 samples were analyzed for *SGLT-1* and *GLUT-2* gene expression, while the expression of *PepT-1* was determined for 56 samples.

### 2.8. Calculations and Statistical Analysis

The data on feed allowance and growth performance generated in the large-scale study were analyzed and evaluated with SEGES Innovation P/S. The growth performance data were statistically analyzed using a Proc Mixed model in SAS (SAS inst. Inc., Cary, NC, USA). Data on the creep feed allowance were not statistically analyzed, as they were based on estimates per double pen. The data were averaged and are presented as simple mean values of allocated dry matter per pig within the treatment groups.

The statistical analysis of BW at weaning and nine weeks of age, average daily gain (ADG) and accumulated gain was based on individual BW measurements taken on the day of weaning and again 4 or 5 weeks after weaning for treatment groups W5 and W4, respectively. The analysis was performed using the following model, where a single pig is the experimental unit:Y_ijk_ = μ + α_i_ + β_j_ + ρ_k_ + (αβ)_ij_ + φ_ij_ + θ + δ_θ_ + ω_ijk_ + ε_ijk_

Y_ijk_ denotes the response variables of BW at weaning, BW at nine weeks of age, ADG and accumulated gain, and μ is the independent variable mean. The systematic effects of WA and FS are denoted as α_i_ (i = W4, W5) and β_j_ (j = DF, LF), respectively. The systematic effect of sow parity is denoted as ρ_k_ (k = 2, …, 8), and the model includes (αβ)_ij_ as the interaction between WA and FS. The number of pigs in a single weaner pen is included as φ_ij_, and θ denotes the random effect of the sow and reflects sow-dependent parameters such as the milk yield and genetics. Litter size is included in the model as δ_θ_, and pig body weight at day three is denoted as ω_ijk_ and included as a covariate. ε_ijk_ is the normally distributed random residual of a model. The data and model residuals were assessed and confirmed to be normally distributed. The least square means for BW at weaning and nine weeks, ADG and accumulated gain were calculated for each treatment group.

The sub-population dataset, including BW at euthanasia, the weight of digestive organs, SI length, organ weight relative to BW, disaccharidase activity and the gene expression of *SGLT-1*, *GLUT-2* and *PepT-1*, contained individual measurements and values for the 60 sacrificed pigs, taking a single pig as the experimental unit for these parameters. The data were analyzed using R 4.1.1 and R studio version 1.4.1717 (R Development Core Team, 2020; The R Foundation, 2021, Indianapolis, IN, USA). The sub-population dataset was analyzed using linear modelling. The models were reduced using AIC-stepwise reduction, determining the significance level of the included systematic effects and their interactions.

The residuals of each model were evaluated using qqnorm to ensure a normal distribution. The Emmeans function was used to obtain least square means, which are presented with the pooled standard error of the mean (SEM). The data analysis was based on the following model:Y_ijk_ = μ + α_i_ + β_j_ + δ_k_ + (αβ)_ij_ + (αδ)_ik_ + (βδ)_jk_ + ε_ijk_,
where Y_ijk_ denotes the result of an individual response variable, and μ is the mean of that variable. The systematic effect of WA (i = W4, W5) is denoted as α_i_, while β_j_ is the systematic effect of FS (j = DF, LF). Here, δ_k_ is the systematic effect of sex (k = female, male). Second-order interactions between the main factors are denoted as (αβ)_ij_, (αδ)_ik_ and (βδ)_jk_, and ε_ijk_ is the normally distributed random residual of a model.

Using AIC-stepwise reduction, the model was reduced to the following:Y_ijk_ = μ + α_i_ + β_j_ + δ_k_ + (αβ)_ij_ + ε_ijk_.

Only the interaction between WA and FS was included in the analysis, as the AIC-stepwise reduction revealed that the other second-order interactions and the third-order interaction were statistically non-significant (*p* > 0.10). Where relevant, Tukey–Kramer adjusted LS means were obtained and presented with the pooled SEM.

## 3. Results

### 3.1. Feed Allocation

The average feed allocation per pig, measured from the time of placement in the weaner unit to nine weeks of age, can be seen in Table 4. Generally, W4 pigs were allocated more DM, as they stayed in the weaner unit for one additional week compared to the W5 pigs. The feed efficiency was numerically higher in the DF groups compared with the LF groups.

### 3.2. Growth Performance

The results for post-weaning growth performance can be seen in Table 5. No overall interactions were found. Generally, both age and LF increased post-weaning growth. Pigs in the W5LF group were the heaviest at weaning and at nine weeks, but they did not have the largest accumulated gain during the weaning period, as pigs from the W4LF group had the largest total gain. The lowest post-weaning ADG was found for the W4DF pigs, followed by those in the W4LF group. Pigs in the W5DF group had an even further increased post-weaning ADG, and the largest ADG was seen in the W5LF pigs.

Considering the effects of the main factors of WA and FS (W4 = DFW4 + LFW4, W5 = DFW5 + LFW5, DF = DFW4 + DFW5, LF = LFW4 + LFW5), the following results are significant. Pigs were heavier in W5 than in W4 (9.35 vs. 7.11 kg BW, *p* ≤ 0.05), and LF pigs were heavier than DF pigs (8.55 vs. 7.91 kg BW, *p* ≤ 0.05) five days after weaning. At nine weeks of age, the DF pigs weighed 20.6 kg and LF pigs weighed 21.8 kg (SE 0.108, *p* < 0.0001), while the W4 pigs weighed 21.0 kg and W5 pigs weighed 21.5 kg (SE 0.108, *p* < 0.01).

### 3.3. Body Weight at Euthanasia and Organ Weights

The body weight at euthanasia and organ results can be seen in Table 6, where the results are presented as LS means for each treatment group. Though sex was included in the model based on the completed AIC model reduction, the analysis showed that sex had no significant effect on the BW and organ response variables when evaluating the generated *p*-values of the final models for these parameters. Therefore, the presented results are averaged over sex, and they relate to WA, FS and their possible interaction.

As hypothesized, W5 pigs were heavier than W4 pigs at euthanasia, and LF pigs were heavier than DF pigs. In accordance with the increased BW, W5 pigs had heavier digestive organs compared with W4 pigs, and similarly, LF pigs had heavier organs than DF pigs. However, when organ weight was considered in relation to BW, only the relative SI weight was significantly affected by WA and FS, as the W5 and LF pigs had an increased relative SI weight compared with W4 and DF pigs, respectively. The length of the SI was significantly affected by WA (*p* = 0.0003) but tended to only be slightly affected by FS (*p* = 0.068). Therefore, W5 pigs had a longer SI than W4 pigs, and LF pigs tended to have a longer SI than DF pigs.

### 3.4. Enzyme Activity

The results for brush border enzyme activity and nutrient transporter gene expression are presented in Table 7. Again, the results are presented as LS means for each treatment group, though the interactions between WA and FS were non-significant for all the response variables, except for a tendency to affect maltase activity. Additionally, the results in Table 7 are averaged over sex, as sex did not interact with WA or FS. Significant effects of sex are reported separately.

Lactase and sucrase activity was unaffected by WA and FS, and the Tukey–Kramer-adjusted means did not differ significantly between the four treatment groups (*p_Lactase_* = 0.164, *p_Sucrase_* = 0.127). Maltase activity was affected by FS (*p* = 0.048). As can be seen in Figure 1, the DF pigs had increased maltase activity compared with pigs receiving LF as a creep feed.

The sex of the pigs was included in the statistical model, but it did not affect maltase or sucrase activity. However, sex did tend to affect lactase activity (*p* = 0.099), as male pigs had a higher activity of 761 U/mg compared to the 582 U/mg lactase activity seen in female pigs.

### 3.5. Gene Expression of Nutrient Transporters

As seen in Table 7, the gene expression levels of *GLUT-2* and *PepT-1* were unaffected by the WA and FS treatments. Additionally, *GLUT-2* was unaffected by sex. However, sex did significantly affect the *PepT-1* gene expression level (*p* = 0.002), as males had a mean ∆Ct value of 7.97, compared to females’ mean ∆Ct value of 6.57.

Like maltase activity, *SGLT-1* gene expression was unaffected by WA. However, the expression of *SGLT-1* was affected by both FS and Sex. Considering the inverted relationship between ∆Ct and the gene expression level, Figure 2a shows that the DF pigs had a decreased expression of *SGLT-1* genes in the SI compared to the LF pigs (*p* = 0.011), and Figure 2b shows that female pigs had increased gene expression compared to male pigs (*p* = 0.003).

## 4. Discussion

The aim of the present study was to investigate the effects of weaning age and the strategy of creep feeding on the digestive and absorptive capacities of newly weaned pigs. It was hypothesized that an additional week of suckling and the allocation of liquid feed would accelerate intestinal maturation and increase the utilization of nutrients in post-weaning pigs. The potential increase in utilization was measured as BW gain and organ weight gain, whereas the brush border activity of disaccharidases and the level of gene expression of selected nutrient transporters were used as indicators of intestinal maturation. The digestion of carbohydrates (CHO) was prioritized above other nutrients, as CHO was the main nutrient in the allocated creep and weaner feeds. Furthermore, the CHO complexes in the feeds would have been distinctly different than those in sow milk. The activities of SI brush border maltase, lactase and sucrase were selected as parameters attesting to the pigs’ digestive capacity for CHO [23,24,25]. Additionally, the *SGLT-1* and *GLUT-2* gene expression levels served as parameters for the absorptive capacity of monosaccharides, and the gene expression level of *PepT-1* was included as a parameter for peptide absorption.

Observing the main effect of the FS, LF increased the BW at nine weeks compared to dry feed by more than one kilo. This was the result of a continuous increase in the BW of LF pigs compared with DF pigs. At weaning, LF pigs were approximately 225 g heavier, while they were 640 g heavier at euthanasia and 1150 g heavier at nine weeks of age. At euthanasia, approximately 70 g of the difference in BW could be explained by an observed increase in the weight of the digestive organs, leaving approximately 570 g of BW unaccounted for. Therefore, this additional weight must have been retained in other organs, as fat, muscle or bone tissue or as water. In a study by Lynegaard et al. [26], 24-day-old weaned pigs retained approximately 7% body fat and 55% muscle tissue. Madsen et al. [27] found that the carcasses of 28-day-old weaned pigs contained 26% DM, and chemical analysis of the DM revealed 14% ash, 27% ether extract and 57% crude protein. Therefore, it seems reasonable to assume that the additional weight of the LF pigs could be explained largely by muscle growth and associated water retention.

As expected, WA affected BW, so that W5 pigs were heavier than W4 pigs at weaning, five days later at euthanasia and at nine weeks of age. At euthanasia, W5 pigs were 2.24 ± 0.19 kg heavier, and collectively, their weighed digestive organs were 170 g heavier than those of W4 pigs. At week nine, W5 pigs were approximately 0.55 ± 0.18 kg heavier than W4 pigs. These findings are partly in accordance with an earlier study by Leliveld et al. [28], where pigs weaned at 5 rather than 4 weeks of age were 2.2 kg heavier at weaning and 3.9 kg heavier 2 weeks post-weaning. However, Leliveld et al. [28] did not find a significant effect of later weaning on BW at 10 weeks of age, arguing that WA accounts for most of the effect on growth performance in the early post-weaning period. This could be a consequence of the growth rate decreasing with age; therefore, W4 pigs would have caught up with W5 pigs in terms of BW gain. However, the effect of WA observed in the later post-weaning period in this study is supported by the study of Faccin et al. [29], who found that the increase in WA from day 19 to 28 linearly increased BW from weaning to 10 weeks of age.

As previously mentioned, the weight of the digestive system differed between pigs weaned in week 4 and 5 and in pigs fed DF compared to those fed LF. As expected, older pigs had heavier stomachs, SIs and colons. Similarly, LF pigs had heavier organs than DF pigs. However, only the relative organ weight of the SI was affected by WA and FS, with older and LF-fed pigs having an increased SI weight in relation to BW. Having a larger and heavier SI is only beneficial when accompanied by increased digestive and absorptive capacities, meaning a lower organ weight relative to BW. In another context, this was also exemplified by high inclusion rates of dietary fiber, leading to increased intestinal weight but lower carcass weight [30]. These conflicting results emphasize the need to link selected biomarkers of gut maturation and the absorptive and digestive capacities of the SI, such as brush border disaccharidase activity and nutrient transporter gene expression, to the more apparent digestibility of carbohydrates.

Contradicting the hypothesis, the present study did not find that increasing WA or feeding on LF increased disaccharidase activity in the proximal SI 5 days post-weaning. In fact, DF significantly increased maltase activity but showed a downregulated expression of *SGLT-1* genes compared to LF-fed pigs. It could be expected that if FS affected maltase activity and *SGLT-1* gene expression, the two parameters would show a positive relation, as increased maltase activity should lead to an increased luminal glucose concentration and might therefore stimulate an increase in *SGLT-1* transporters in the brush border, thus optimizing feed utilization. However, the present results do not indicate such a relation.

It can be speculated that LF might stimulate additional *SGLT-1* transporters to be formed in the intestinal brush border [31]. As the DF and LF groups had numerically similar DM allocations, it was assumed that they consumed similar amounts of DM, and therefore, the heavier LF pigs had greater feed utilization, as reflected in the results for feed efficiency. Consequently, the present study cannot confirm that an elevated level of enzyme activity results in the increased utilization of nutrients, though the results may indicate that feed utilization depends more on the absorptive capacity of the SI than on the digestive capacity.

Though other studies have found that the level of brush border enzyme activity and nutrient transporter gene expression are indicative of gut maturation in piglets [23,24,25], it is questionable whether enzyme activity and the nutrient transporter gene expression level are suitable parameters when applied to gut tissues from post-weaned pigs reared under commercial conditions. Another parameter of interest when examining gut maturation is gut integrity, assessed by measuring villus height, crypt depth and digestibility. Higher villi increase the surface area available for nutrient absorption and are thus considered as an indirect indicator of intestinal digestive capacity. Jijang et al. [32] found that 24-day-old weaned pigs receiving LF had the same levels of maltase and sucrase activity as pigs given the same diet as DF; however, the LF pigs had a significantly increased jejunal villus height and a higher apparent digestibility of dietary fat and ash. This underlines the fact that there is not necessarily a direct link between the activity of disaccharidases and gut morphology, and even more curiously, there is no immediate link between the activity of disaccharidases and CHO digestibility. Furthermore, villus height and crypt depth are dynamic markers, in which significant changes can occur within a few days, even within the same treatment group [33]. Again, this all emphasizes the complexity of the digestive process and stresses the difficulties in selecting suitable markers for gut maturation as well as digestive capacity.

Discussing the increased maltase activity seen in the DF pigs, it might be argued that DF strains the gut more than LF, which could result in a decrease in *SGLT-1* expression but an upregulation in maltase activity to compensate. Indeed, this could attest to the adaptability of the SI, but it is questionable whether this is actually a compensating mechanism or a symptom of an exhausted organ in energy deficit. A previous study by Byrgesen et al. [17] showed results similar to those of the present study, where maltase, sucrase and lactase activities were elevated in dry-creep-fed pigs at the age of 25 days, just prior to being weaned. The authors argued that the amount of DM substrate itself affects brush border disaccharidase activity through a downstream relationship with salivary amylase [17]. In the current study, LF pigs were heavier and had upregulated *SGLT-1* function, but DF pigs had increased maltase activity; thus, it is questionable whether enzyme activity and nutrient transporter gene expression are suitable measures of the digestive utilization of feed, as these measures do not correspond with the growth performance results. It should also be noted that the increase in maltase activity could be due to differences in nutrient composition between the two groups, although this was only the case for a short period during pre-weaning. Future research should consider the possible correlations between enzyme activity, nutrient digestibility and gut morphology.

In the pre-weaning study presented in Lyderik et al. [19], very few differences were observed between the groups, indicating that it is perhaps necessary for weaning to take place in order to observe differences in gut absorption and function. This seems to be the case, as FS had WA-independent effects on BW at weaning, euthanasia and nine weeks of age, as well as organ weight, the relative weight of the SI, maltase activity and the expression of *SGLT-1* genes.

## 5. Conclusions

The provision of liquid rather than dry creep feed before weaning and during the early post-weaning period resulted in heavier pigs at weaning and at nine weeks of age. Similarly, a five- rather than four-week weaning age also increased body weight at nine weeks after weaning. Unexpectedly, the increased growth of pigs fed on liquid feed around weaning was not associated with an increase of disaccharidase activity in the proximal small intestine during the early post-weaning period but was associated with an upregulation in the expression of glucose transporter genes (*SGLT-1*).

## Figures and Tables

**Figure 1 animals-13-01851-f001:**
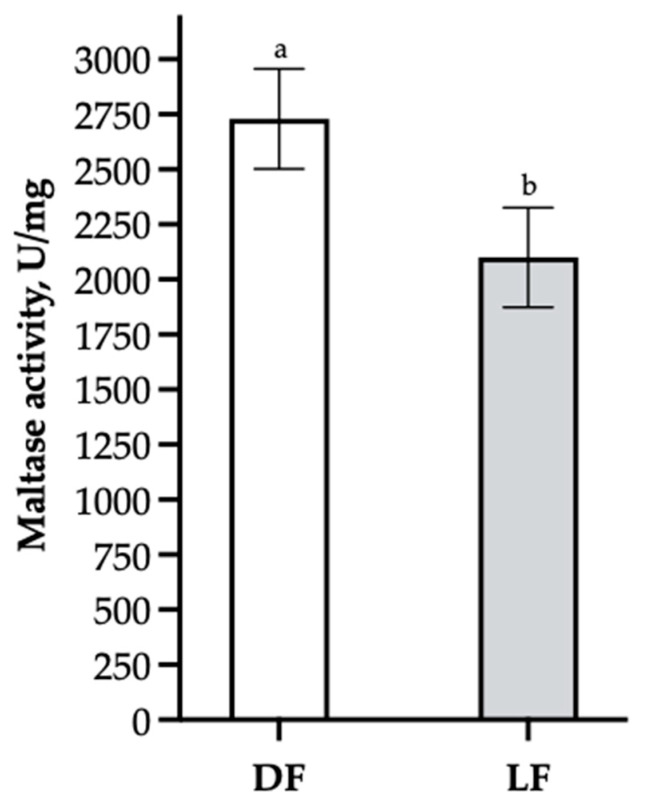
LS means of maltase activity and pooled SEM showing the effect of feeding strategy (FS). Pigs received either dry (DF) or liquid feed (LF) as a creep feed. Pigs were sampled 5 days post-weaning. ^a,b^ Columns without a common superscript differ significantly (*p* = 0.048).

**Figure 2 animals-13-01851-f002:**
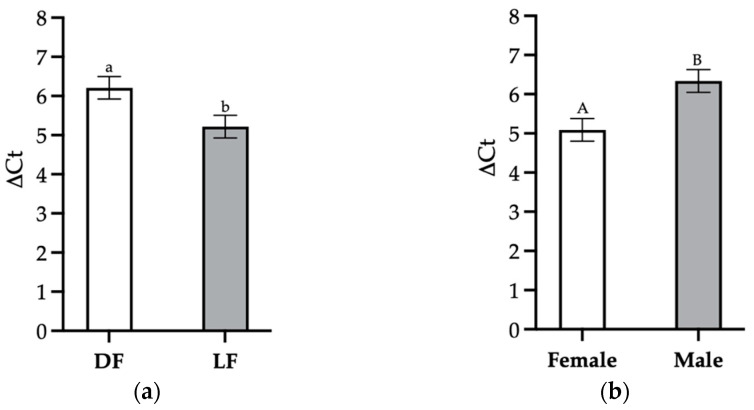
Effects of feed strategy (FS) and sex on the *SGLT-1* gene expression level in the small intestines of weaned pigs at 5 days post-weaning. Pigs received either dry (DF) or liquid (LF) creep feed prior to weaning. Results are reported as ∆Ct values; thus, the negative relationship between ∆Ct and the gene expression level should be considered during interpretation. (**a**) Mean ∆Ct values for *SGLT-1* in DF and LF pigs. ^a,b^ Columns without a common superscript differ significantly (*p* = 0.011). (**b**) Mean ∆Ct values for *SGLT-1* in female and male pigs. ^A,B^ Columns without a common superscript differ significantly (*p* = 0.003).

**Table 1 animals-13-01851-t001:** Ingredients and compositions of milk supplements in their dry form.

Ingredients ^1^	Milk Supplement	
	MS1 ^2^	MS2 ^3^
Dairy biproducts	X	X
Whey protein concentrate	X	X
Whey powder	X	X
Heat-treated wheat starch		X
Vegetable fat		X
Vegetable protein		X
Chemical analysis (dry matter basis) ^4^		
Energy, MJ/kg DM	20.9	15.2
Dry matter, %	97.0	94.7
Crude protein, %	19.8	22.1
Crude fat, %	16.8	12.2
Ash, %	5.0	3.3
Lysine, %	1.81	1.70
Threonine, %	1.29	1.07
Methionine, %	0.41	0.50
Valine, %	1.19	1.21

^1^ As the milk supplements used were commercial products, the lists of ingredients were known. However, the exact amounts included and the ratios between them were undisclosed. X indicates whether an ingredient was included in the milk supplement. ^2^ MS1 denotes Agilia™ Danmilk™ Supreme 1.0. Piglets in all treatment groups had access to MS1 from 24 h post-partum to day 8. ^3^ MS2 denotes Agilia™ Danmilk™ Gain. Piglets in LF treatment groups had access to MS2 from post-partum days 9 to 17. ^4^ Values from the chemical analysis performed with SEGES Innovation P/S.

**Table 2 animals-13-01851-t002:** Compositions of weaner diets ^1^ in their dry form.

Weaner Diet 1 (6–9 kg) Creep Feed		Weaner Diet 2 (9–18 kg)		Weaner Diet 3 (>18 kg)	
Ingredients					
Wheat, %	47	Wheat, %	52	Wheat, %	35
Barley, %	13	Barley, %	9	Barley, %	12
Rye, %	8	Oats, %	2	Oats, %	1
Fishmeal, %	9	Potato, %	12	Potato, %	31
Pig lard, %	3	Fishmeal, %	3	Pig lard, %	<1
Concentrate ^2^, %	20	Pig lard, %	1	Soybean meal, %	16
		Soybean meal, %	13	Mineral supplement ^3^, %	4
		Fava beans, %	3		
		Mineral supplement ^3^, %	5		
Chemical composition ^4^		Calculated composition ^5^		Calculated composition ^5^	
Dry matter, %	89.4		78.2		63.2
Net energy, MJ/kg	10.5		2.8		2.8
Crude protein, % DM	18.8		4.5		3.7
Crude fat, % DM	6.9		0.9		0.6
Ash, % DM	4.7		1.4		1.2
Zinc, % DM	0		0		0
Lysine, % DM	1.48		0.35		0.29
Threonine, % DM	0.92		0.21		0.18
Methionine, % DM	0.48		0.10		0.08
Valine, % DM	0.92		0.20		0.18

^1^ The weaner diets were mixed on-farm and optimized according to the current norms set by SEGES Innovation P/S [20]. ^2^ A commercial concentrate with a known list of ingredients but undisclosed ratios between them. The ingredients are soy protein concentrate, lactose, toasted soybeans, wheat, mono-dicalcium-phosphate, shea and palm fat, saccharose, natrium chloride, soy oil, barley, wheat bran, calcium carbonate, wheat, maize, water-soluble onion residues and water-soluble grape seed residues. ^3^ A commercial mineral product with an undisclosed composition. ^4^ Chemical analysis of weaner diet 1 was performed with SEGES Innovation P/S. ^5^ The calculated compositions of weaner diets 2 and 3 were provided by the owner of the herd.

**Table 3 animals-13-01851-t003:** Forward and reverse primers for the selected nutrient transporter genes and reference gene.

Transporter	Forward Primer (From 5′ to 3′)	Reverse Primer (From 5′ to 3′)
*SGLT-1*	AATGCGGCTGACATCTCTGT	CCAACGGTCCCACGATTAGT
*GLUT-2*	AGTTGGCGCTATCAACACGA	CACAAGTCCCACCGACATGA
*PepT-1*	CAGACTTCGACCACAACGGA	TTATCCCGCCAGTACCCAGA
*B2M*	CAAGATAGTTAAGTGGGATCG	TGGTAACATCAATACGATTTC

**Table 4 animals-13-01851-t004:** Average feed allocation per pig from placement in the weaner unit to nine weeks of age ^1^.

WA	W4		W5	
FS	DF	LF	DF	LF
Weaner diet 1, kg/pig	6.2	6.1	5.6	6.7
Weaner diet 2, kg/pig	13.4	12.2	10.7	10.9
Dry matter ^2^, kg/pig	20.2	19.2	18.6	18.7
Feed efficiency kg/kg growth	1.39	1.28	1.40	1.34

^1^ Abbreviations: WA = weaning age, FS = feeding strategy, 4W = weaning in week 4, 5W = weaning in week 5, DF = dry feed, LF = liquid feed. ^2^ Allocated dry matter from weaning to nine weeks of age.

**Table 5 animals-13-01851-t005:** Results for post-weaning growth performance from weaning to nine weeks of age ^1^.

WA	W4		W5			*p*-Value	
FS	DF	LF	DF	LF	SEM	WA	FS
Number of pigs	3106	3179	3013	3171			
Weaning weight, kg	6.15	6.37	7.87	8.10	0.05	<0.0001	<0.0001
Weight at week 9, kg	22.0	23.2	22.6	23.7	0.18	0.014	<0.0001
Accumulated gain, kg	15.9	16.8	14.6	15.6	0.15	<0.0001	<0.0001
ADG, g/day	407	434	459	486	4.3	<0.0001	<0.0001

^1^ Abbreviations: WA = weaning age, FS = feeding strategy, 4W = weaning in week 4, 5W = weaning in week 5, DF = dry feed, LF = liquid feed.

**Table 6 animals-13-01851-t006:** Results for BW at euthanasia and organ parameters presented as LS means ^1^ for each treatment group ^2^.

		W4		W5			*p*-Value		
		DF	LF	DF	LF	SEM	WA	FS	WA × FS
	No. pigs	15	15	15	15				
	BW at euthanasia ^3^, kg	6.93 ^a^	7.28 ^a^	8.88 ^b^	9.82 ^b^	0.270	<0.0001	0.017	0.262
Organ weight, g	Stomach	42.0 ^a^	47.3 ^a^	60.2 ^b^	63.5 ^b^	1.98	<0.0001	0.030	0.606
	Small intestine	260 ^a^	296 ^a^	366 ^b^	440 ^c^	17.0	<0.0001	0.002	0.260
	Colon	107 ^a^	111 ^ab^	129 ^bc^	148 ^c^	5.32	<0.0001	0.033	0.158
Organ length, m	Small intestine	9.34 ^a^	9.75 ^a^	10.27 ^ab^	10.87 ^b^	0.275	0.0003	0.068	0.711
Relative organ weight ^4^, g/kg	Stomach	6.11	6.6	6.76	6.47	0.214	0.246	0.654	0.067
	Small intestine	37.5 ^a^	41.0 ^ab^	41.0 ^ab^	45.0 ^b^	1.73	0.025	0.029	0.912
	Colon	14.2	15.3	14.5	15.2	0.787	0.942	0.300	0.802

^1^ Group means are Tukey–Kramer-adjusted and presented with pooled SEM values. The means are averaged over sex, as sex was non-significant for the response variables reported here. ^2^ Abbreviations: WA = weaning age, FS = feeding strategy, 4W = weaning in week 4, 5W = weaning in week 5, DF = dry feed, LF = liquid feed. ^3^ Pigs were euthanized 5 days post-weaning. ^4^ Organ wight (g) relative to body weight (kg). ^a,b,c^ LS means without a common superscript differ significantly.

**Table 7 animals-13-01851-t007:** Results for brush border disaccharidase activity and the gene expression of nutrient transporters in the SI of weaned pigs 5 days post-weaning. Results presented as LS means ^1^ for each treatment group ^2^.

		W4		W5			*p*-Value		
		DF	LF	DF	LF	SEM	WA	FS	WA × FS
Enzyme activity, U/mg	Maltase	2170	1701	2884	2890	324	0.637	0.048	0.094
	Lactase	658	539	839	655	160	0.360	0.808	0.164
	Sucrase	734	962	560	854	125	0.164	0.297	0.127
Gene expression, ∆Ct	*SGLT-1*	5.14	5.97	6.09	5.59	0.47	0.618	0.011	0.800
	*GLUT-2*	7.35	7.14	7.86	7.16	0.41	0.742	0.141	0.231
	*PepT-1*	6.88	7.45	8.26	6.60	0.43	0.988	0.873	0.307

^1^ Group means are Tukey–Kramer-adjusted and presented with pooled SEM values. The means are averaged over sex, as interactions of sex with WA and FS were excluded from the statistical model, as stated in the Section 2. The effect of sex is reported separately. ^2^ Abbreviations: WA = weaning age, FS = feeding strategy, 4W = weaning in week 4, 5W = weaning in week 5, DF = dry feed, LF = liquid feed, *SGLT1 =* sodium-glucose cotransporter-1, *GLUT2* = glucose transporter-2, *PepT1* = peptide transporter-1.

## Data Availability

Data is unavailable due to privacy restrictions.

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
