# Peer review of "Effects of Weaning Age and Creep Feed Type on Growth Performance and Gut Maturation in Weaned Piglets"

_animals, 2023, doi:10.3390/ani13111851_

Round 1

Reviewer 1 Report

The concept of this study is very important in the swine industry, however experimental setting was ambiguous even if this study was conducted in the clinical condition. 

The intestine was developed day by day in the pre- and post-weaning period, your group setting (4w vs 5w) was too ambiguous to compare the groups. 

Selection criteria for sacrificed piglets was obscure (the authors only described sacrificed piglets were selected by the BW and sex). Not only the ages but also the stocking density were influenced the function of the small intestine. 

Small intestine was a long organ, therefore the authors must mention the sampling point of the small intestine strictly. The function was not same among the “proximal SI”. In addition, the tissue was not wash before the sampling? 

Discussion was poor. Many sentences just summarize the results.

Author Response

Rebuttal for Manuscript ID: animals-2326726

This cover contains quotations from the three received review-reports, and the authors’ rebuttal for each comment or suggested alteration. The rebuttal is highlighted as green script.

Review Report 1

“The concept of this study is very important in the swine industry. However, experimental setting was ambiguous even if this study was conducted in the clinical condition.”

The authors would like to thank Reviewer 1 for their useful comments and suggestions.

We do acknowledge that the experimental setup has its limitations, given the fact that it was conducted under commercial conditions. However, it was in our interest to document possible effect of weaning age and creep on gut maturation, that has previously been found under clinical conditions, in settings of a standard production herd. .

“The intestine was developed day by day in the pre- and post-weaning period, your group setting (4w vs 5w) was too ambiguous to compare the groups.”

We acknowledge, that one week apart may have to small of an age difference to generate notable differences. However, a one-week extended lactation period was chosen, as this would be a realistic and implementable strategy for many producers. Extending the lactation even further would be too costly for many producers compared to the benefits of the strategy.  This argument has been added to 2. Method and Materials.

“Selection criteria for sacrificed piglets was obscure (the authors only described sacrificed piglets were selected by the BW and sex). Not only the ages but also the stocking density were influenced the function of the small intestine.”

Selection criteria has been clarified under 2.5 Organ and tissue sampling. Stocking density was not included as a selection criterion, as practical on-farm conditions did not allow us to consider it. This too is now clarified.

“Small intestine was a long organ, therefore the authors must mention the sampling point of the small intestine strictly. The function was not same among the “proximal SI”. In addition, the tissue was not wash before the sampling?”

We completely agree with this comment, and this has been rectified under 2.5 Organ and tissue sampling. The tissue was rinsed with water, and this too is now mentioned in the paragraph.

“Discussion was poor. Many sentences just summarize the results.”

Thank you for this comment. The point has been considered carefully. We prefer to write in a concise and clear manner, and refrain from excessive use of speculations. For this reason, we have kept the discussion short and only commented on the actual findings of this study. We sincerely hope this can be accepted.   

Reviewer 2 Report

It is significant that the manuscript concerning a practical aspect of piglets in terms of weaning age and creep feed type. However, numbers of pigs used and sampling time should be checked, and other modifications and errors correction are needed to a publishable standard.

Major comments as follows:

Page 1 line 31 and Page 3 line 93, 12,469 pigs vs. 12,932 piglets? Why?

Page 1 line 33, Page 3 line 98 and 109, piglets euthanized five days after weaning vs. pigs were sacrificed 3 days before weaning, which one is correct, or what is the fact?

Page 4 line 145-150, performance difference between DF and LF groups may be attributed to different diets from 9 to 17 after weaning, since DF treatments and LF groups were provided with different diets with different nutrients (dry weaner diet vs. MS2) during this period, please explain it.

Page 5 line 171 Table 2, “weaner diet 2 (<18kg)” should be “… (> 18kg)”? Chemical composition was missed for diet 2 in Table 2.

Page 5 line 182, the sampling time need to be addressed in detail, and there were 28 (16+12) piglets sacrificed, which is inconsistent with the number of sixty in “Animals and experimental design” part, please check it.

Page 8 line 307-309, the content in the Results was different from that in Table 5, please explain it.

Page 10 line 376-379, It was opposite that authors addressed that DF pigs decreased SGLT-1 gene in the SI, and female increased gene expression, however, the fact was that DF pigs increased SGLT-1 gene in the SI, and female decreased gene expression in Figure 2 a and b, please check and rewrite.

Reference: please correct the style of reference following the guide for author of Animals, for example, the names of cited journal in full or abbreviation? and so on.

Author Response

Review Report 2

”It is significant that the manuscript concerning a practical aspect of piglets in terms of weaning age and creep feed type. However, numbers of pigs used and sampling time should be checked, and other modifications and errors correction are needed to a publishable standard.”

The authors thank Reviewer 2 for the comments and corrections. These are highly relevant and are now rectified.

”Page 1 line 31 and Page 3 line 93, 12,469 pigs vs. 12,932 piglets? Why?”

This is an error. Both should be 12,932. This is now rectified.

”Page 1 line 33, Page 3 line 98 and 109, piglets euthanized five days after weaning vs. pigs were sacrificed 3 days before weaning, which one is correct, or what is the fact?”

This comment refers to the second paragraph of the section 2.2 Animals and experimental design. Authors have attempted to explain that from the large-scale study two sub-populations were sampled – one sub-population consisted of 60 pre-weaning piglets (euthanised and sampled 3 days prior to weaning) and the other of 60 post-weaning piglets (euthanised and sampled 5 days after weaning). The results of the sub-populations are reported separately, and the current article reports the findings relating to the post-weaning sub-population. The written paragraph refers to the presentation of pre-weaning results as Lyderik et al. (in press), and additionally, it is clarified by the sentence: “The subpopulation described in the present study, consisted of 60 post-weaning pigs euthanized in the immediate post-weaning period.” (L. 117).

We acknowledge that this could be explain some of the difference. A sentence in the discussion has now been included related to the difference Maltase activity.

Table 2 has now been rectified, and include information about all three post-weaning diets. We apologize very much for the confusion and missing data.

”Page 5 line 182, the sampling time need to be addressed in detail, and there were 28 (16+12) piglets sacrificed, which is inconsistent with the number of sixty in “Animals and experimental design” part, please check it.”

We understand the confusion and have attempted to clarify, that for each of the first 3 weeks 16 piglets were sacrificed, and the last week 12 were sacrificed. The math is then (3 x 16 pigs) + (1 x 12 pigs) = 60 pigs. Hopefully this is now understandable.

We have now tried to clarify sampling time as best possible without adding dates and the exact time of day. Alterations can be seen in the first paragraph of 2.5 Organ and tissue sampling.

”Page 8 line 307-309, the content in the Results was different from that in Table 5, please explain it.”

The results in L. 307-309 are pooled results reporting main effects of WA and FS, which has now been clarified in the text. However, we understand the confusion, as the table references should have been deleted, because we have changed these tables. The problem has been rectified.

”Page 10 line 376-379, It was opposite that authors addressed that DF pigs decreased SGLT-1 gene in the SI, and female increased gene expression, however, the fact was that DF pigs increased SGLT-1 gene in the SI, and female decreased gene expression in Figure 2 a and b, please check and rewrite.”

The following has been added to 2.7 Analysis of nutrient transporter gene expression: ”When interpreting the ∆Ct results, it is important to consider the inverted relation between ∆Ct and the gene expression level, where a lower ∆Ct-value signifies a higher level of gene expression.”. Additionally, the inverted relation is now mentioned again in the result section 3.5. and a third time in the figure text.

”Reference: please correct the style of reference following the guide for author of Animals, for example, the names of cited journal in full or abbreviation? and so on.”

The reference list has been reviewed and should now comply with the official guide for authors. The names of cited journal have now been changed to the correct abbreviations.

Vend tilbage til dette efter møde

Menes der noget ala “additional results and discussion” i mit speciale?

Reviewer 3 Report

This study “Effects of weaning age and creep feed type on growth performance and gut maturation in weaned piglets” is a nice study with a large number of pigs (large-scaled experiment). The results are meaningful and of significance to swine researchers and producers. However, the manuscript is very hard to follow and need to be revised, and the data presentation is not clear and need to be clarified.    

Comments:

In abstract, please use p < 0.01/0.05) such as Line 35, use  P < 0.05; Line 36 use p < 0.01, and Line 39 use p < 0.01

In material and methods, part 2.2, the experiment lasted one year, was there any environmental (seasonal) effect ---block design? Why was the period 8 weeks + 4days? The authors need to specify the selection standard for the subpopulations. Was the selection randomly or based average weight within the group? For the names of the groups, WK4 and WK5 are a bit confusing to the reader. I would prefer if days were used like D28 or D35 OR of if something like “early weaned” or “late weaned” was used. Specify the reason for the sixty piglets for being sacrificed 3 days prior to being weaned, specify the treatment, sex and average weight and the season; Was 15 pigs/group? Similarly, specify the time of post weaning for the 60 post-weaning pigs, specify the treatment, sex, average weight and the season. Was 15 /group? Are these two group from the same sows?

Part 2.4, Is there any reason for not reporting the percentage of each ingredient in MS1 and 2 (table 1)? A “fatal flaw” issue is that diet form is construed with diet composition. If we wanted to test if there was a difference between liquid vs. dry, both the dry and liquid feeds should be the same composition, but that is not the case. Between D9 and D17 the DF diet gets the weaner diet 1, whereas the LF pigs get MS2. While one is dry and the other is liquid, they are vastly different in their composition. I would have left this week long phase out of the study since from D18 on both the DF and LF are getting the same weaner 1 creep diet in dry or liquid form.

One question regarding the post-weaner diet 2, why is it that the DF pigs received liquid feed in the post-weaning phase (line 164-165). Was there a reason for this change from a dry feed post-weaning to a liquid feed? Or is this just common practice?

One smaller issue is that it’s confusing to the reader if the time is pre or post-weaning. For example, from line 145 to 150, its stated in days postnatal of when feed is being given. However, in line 163-165 there’s mention of “first 8 days”. It would be clearer if there was a direct timeline from postnatal to end of study days are used. For example, it should be a timeline of Day 0 to D50 something or whenever the pigs reached 18 kg.

2.8. For your statistical analysis on line 250, it’s stated that a single pig is the experimental unit. For the intestinal measurements this is expected since there are a total of 60 pigs (n = 60). However, for the pig performance, pen should be the experimental unit, not a single pig.

3.3. This table (Table 6) can be reduced to just relative organ weight. A bigger pig is expected to have a bigger stomach, small intestine, etc. It is the relative nature of the organ to the BW that is the important part.

In discussion:  426-443, Could the difference in maltase activity be because of the different nutrient compositions of the diets? The liquid fed pigs got different nutrients in their MS2 diet, which the DF pigs never got.

Author Response

Review Report 3

“This study “Effects of weaning age and creep feed type on growth performance and gut maturation in weaned piglets” is a nice study with a large number of pigs (large-scaled experiment). The results are meaningful and of significance to swine researchers and producers. However, the manuscript is very hard to follow and need to be revised, and the data presentation is not clear and need to be clarified.”

The authors thank Reviewer for their time and helpful comments.

“In abstract, please use p < 0.01/0.05) such as Line 35, use P < 0.05; Line 36 use p < 0.01, and Line 39 use p < 0.01.”

This has now been altered, though for line 36 and 39 P = 0.01. Both P-values are slightly higher than 0.01, and can therefore not be presented as <0.01, but using one less decimal leave them at P=0.01.

“In material and methods, part 2.2, the experiment lasted one year, was there any environmental (seasonal) effect ---block design? Why was the period 8 weeks + 4days? The authors need to specify the selection standard for the subpopulations. Was the selection randomly or based average weight within the group? For the names of the groups, WK4 and WK5 are a bit confusing to the reader. I would prefer if days were used like D28 or D35 OR of if something like “early weaned” or “late weaned” was used. Specify the reason for the sixty piglets for being sacrificed 3 days prior to being weaned, specify the treatment, sex and average weight and the season; Was 15 pigs/group? Similarly, specify the time of post weaning for the 60 post-weaning pigs, specify the treatment, sex, average weight, and the season. Was 15 /group? Are these two group from the same sows.”

  • Seasonal effect:
    • See earlier statement rebutting Reviewer 1.
  • Period of 8 weeks + 4days:
    •  
  • Selection of animals:
    • The selection criteria have now been clarified under 2.5 Organ and tissue sampling.
  • W4 & W5:
    • We did not wish to call the groups “early” and “late”, as weaning in 4th week is not considered early weaning. Furthermore, we chose to call it “Week 4” and “Week 5”, because these are terms commonly used when speaking of weaning from a management point of view.
    • The fact that performance data needed to be curated for ~13,000 at an independent functioning production site made it impossible to wean on the exact same day of age for every batch and piglet within it. Naming the groups d28 and d35 would be misleading.
  • Pre-weaning pigs:
    • The pre-weaning pigs are not a part of sub-population for which results are reported here. Pre-weaning results are reported in a separate article – Lyderik et al. (in press). As mentioned, when rebutting Reviewer 2, hopefully this has now been sufficiently clarified.
  • Post-weaning pigs:
    • Treatment and number of pigs per group is stated in paragraph 3 of section 2.2. Animals and experimental design. The actual feed details are clarified in 2.4. Diet and feeding regime.

“Part 2.4, Is there any reason for not reporting the percentage of each ingredient in MS1 and 2 (table 1)? A “fatal flaw” issue is that diet form is construed with diet composition. If we wanted to test if there was a difference between liquid vs. dry, both the dry and liquid feeds should be the same composition, but that is not the case. Between D9 and D17 the DF diet gets the weaner diet 1, whereas the LF pigs get MS2. While one is dry and the other is liquid, they are vastly different in their composition. I would have left this week long phase out of the study since from D18 on both the DF and LF are getting the same weaner 1 creep diet in dry or liquid form.”

We acknowledge the confusions that may arise considering the experimental design, namely related to the diet shifts. First of all, Table 2 has now been rectified, and include information about all three post-weaning diets. We apologize very much for the confusion and missing data. However, considering the study was conducted under practical conditions, we had to adapt to the standard procedures of the herd.

“One question regarding the post-weaner diet 2, why is it that the DF pigs received liquid feed in the post-weaning phase (line 164-165). Was there a reason for this change from a dry feed post- weaning to a liquid feed? Or is this just common practice?”

Yes, liquid feeding in weaner and finisher units is very common in Denmark. It is common to provide dry creep feed and then have an immediate change to liquid feed in the weaner section.

As previously mentioned, considering the study was conducted under practical conditions, we had to adapt to the standard procedures of the herd.. Thus, the experiment was designed to accommodate the production routines of the herd, and to not influence the productivity negatively.

“One smaller issue is, that it’s confusing to the reader if the time is pre or post-weaning. For example, from line 145 to 150, its stated in days postnatal of when feed is being given. However, in line 163-165 there’s mention of “first 8 days”. It would be clearer if there was a direct timeline from postnatal to end of study days are used. For example, it should be a timeline of Day 0 to D50 something or whenever the pigs reached 18 kg.”

Thank you for your suggestion. We have not used the “Day 0” or “D50” nomenclature, but we have added clarifying sentences to the description of diet and feeding regime. The relevant paragraphs now start with: “Regarding pre-weaning creep feeding...” and ” As for the diet and feeding regime post-weaning...”. Hopefully this should help readers understand the timelines.

“2.8. For your statistical analysis on line 250, it’s stated that a single pig is the experimental unit. For the intestinal measurements this is expected since there are a total of 60 pigs (n = 60). However, for the pig performance, pen should be the experimental unit, not a single pig.”

Reviewer 3 references to the second paragraph of 2.8. Calculations and statistical analysis. We believe the experimental unit here should be a single pig, because all pigs were weighted individually and not at pen level, as is stated in the paragraph.

“3.3. This table (Table 6) can be reduced to just relative organ weight. A bigger pig is expected to have a bigger stomach, small intestine, etc. It is the relative nature of the organ to the BW that is the important part.”

We agree on the notion to normally only present relatively organ weights and lengths. However, in the case of this study, where absolute small intestine length did in fact differ between feed strategies and not only age, we considered it more correct to also present these values. We hope this can be accepted.

“In discussion: 426-443, Could the difference in maltase activity be because of the different nutrient compositions of the diets? The liquid fed pigs got different nutrients in their MS2 diet, which the DF pigs never got.”

Thank you for this valuable comment. We have now included a sentence in the Discussion; “It should also be noted that the increase in maltase activity, could be due to difference in nutrient composition between the two groups, although this was only for a shorter period during pre-weaning.”

Round 2

Reviewer 1 Report

> We completely agree with this comment, and this has been rectified under 2.5 Organ and tissue sampling. The tissue was rinsed with water, and this too is now mentioned in the paragraph.

Thank you for your response. Please mention why was the authors choose this sampling point. Digestive enzyme activity and absorption of the nutrient was not homogenous through the small intestine, only one point evaluation was not satisfy your aims, I think.

I feel very disappointed that the authors only measured the digestive and absorption activities of the small intestine. Many of previous reports were measured not only the digestive and absorption activities but also the intestinal morphology such as villous height as you wrote. Furthermore, the authors also wrote stress may act the important role for gut maturation. Why do the authors perform the additional analyses about the stress parameters? 

Discussion is still incomplete.  

The authors were not discuss about the organ weight.

Appropriate reference was lack. 

L215: The authors determined the “mucosal” enzyme activity, however mucosal collection methods were not mentioned in the manuscript.

Author Response

Second Rebuttal for Manuscript ID: animals-2326726

This cover contains quotations from the latest review-reports, and the authors’ rebuttal for each comment or suggested alteration. The rebuttal is highlighted as green script.

Reviewer 1

“Please mention why was the authors choose this sampling point. Digestive enzyme activity and absorption of the nutrient was not homogenous through the small intestine, only one point evaluation was not satisfy your aims, I think.”

The authors acknowledge that only analysing samples from the proximal small intestine (SI) may have left out the possibility of identifying treatment effects in other parts of SI undetected. However, we chose to focus on the proximal part of the intestine based on our previous findings published in Amdi et al. (2021)1, where feeding pre-weaning piglets increasing amounts of complex carbohydrates (wheat) in milk replacer affected the maltase and sucrase activity almost exclusively in the proximal intestine. This argument has now been added to the manuscript.

“I feel very disappointed that the authors only measured the digestive and absorption activities of the small intestine. Many of previous reports were measured not only the digestive and absorption activities but also the intestinal morphology such as villous height as you wrote. Furthermore, the authors also wrote stress may act the important role for gut maturation. Why do the authors perform the additional analyses about the stress parameters?”

We do realize, that including an analysis of the SI’s morphological characteristics might potentially show treatment-related differences in the intestines of the sampled pigs. However, due to limited resources, we carefully selected which markers to use as indicators for intestinal maturation in this on-farm trial. As we did our preparation research, we investigated villus hight and crypt depth, and found that these markers could be highly dynamic, as they change rapidly over short periods of time. As an example, D’Inca et al. (2010) found that villus hight and crypt depth changed significantly in two days in new-born piglets. We recognise that the morphological development of the intestine is likely most rapid in the early days post-partum, however, the rapidness of the changes still made us hesitate to use these markers in this on-farm trial, as we could not standardize time of birth in a trial of this size (number of pigs). Moreover, we prioritized the usage of markers with a direct indication of digestive and absorptive capacity.

In regard to stress – our intention was to not mention stress as a factor. We apologize for the confusion. We have used the word “stress” in relation to the intestine in the following sentence: “It is possible that DF compared with LF stresses the gut more and that results in a decrease in SGLT-1 but might in turn upregulate maltase activity to compensate.” Here, we are not referring to the stress-level of a pig, but rather the strain of adaption their intestine might be experiencing after weaning. To avoid any confusion, we have rephrased this argument in the discussion; “When discussing the increased maltase activity seen in DF pigs, it might be, that It is possible that DF  strains the gut more than compared with LF, which could stresses the gut  more and that  results in a decrease in SGLT-1 expression, but an might in turn up-regulates in maltase activity to compensate.” We hope this is acceptable.

Generally, Reviewer 1 has suggested the inclusion of more parameters in the experimental design. Morphology and stress-related parameters has been mentioned. We acknowledge that doing exploratory research demands the inclusion of multiple parameters in order to make valid and evidence-based conclusions. We also acknowledge, that including stress-related parameters would be interesting, as they might elucidate how feeding strategy and weaning age is correlated to stress level at weaning. However, our research group try to focus on specific hypotheses. We did not wish to include several markers hoping they might result in something significant. Additionally, we must again emphasise that the amount of resources dictates research prioritization and conditions.

“Discussion is still incomplete.
The authors were not discuss about the organ weight. Appropriate reference was lack.”

In regard to the discussion, the authors acknowledge that we mention “results” when discussing body composition of pigs and the retention of the additional weight seen in LF and W5 pigs. As the interaction of WAxFS was non-significant for all organ parameters (though one tendency was found), we aim to discuss W4 vs. W5 and LF vs. DF (main effects) and have therefore presented some results in the relevant context in the discussion.  In section “3. Results” we present results as group means in accordance with the experimental setup, so these cannot be referenced when discussing the relation between main effects and retention. We hope this is acceptable and can now be understood from the text.

We have added some discussion of organ weight and relative organ weight. We hope it is satisfactory.

“L215: The authors determined the “mucosal” enzyme activity, however mucosal collection methods were not mentioned in the manuscript.”

We thank Reviewer 1 for this comment. We did not collect mucosal samples. Instead, the analysis of enzyme activity was performed on whole SI tissue samples. Hopefully this is now clear when reading the text.

References used in rebuttal

1Amdi, C.; Pedersen, M.L.M.; Klaaborg, J.; Myhill, L.J.; Engelsmann, M.N.; Williams, A.R.; Thymann, T. Pre-weaning adaptation responses in piglets fed milk replacer with gradually increasing amounts of wheat. Br. J. Nutr. 2021, 126, 375-382, doi:10.1017/S0007114520004225.

2D’Inca, R.; Che, L.; Thymann, T.; Sangild, P.T.; Le Huërou-Luron, I. Intrauterine growth restriction reduces intestinal structure and modifies the response to colostrum in preterm and term piglets. Livestock Science, 2010, 133, 20-22. doi: 10.1016/j.livsci.2010.06.015.

Reviewer 2

Reviewer 2 did not provide further comments or suggestions but emphasised the need for proofreading.

The authors agree. There were several typing mistakes and incomplete restructuring of sentences. Hopefully the manuscript is now at an acceptable level.

Reviewer 2 Report

This revised manuscript can be accepted after error checking.

Author Response

The authors agree. There were several typing mistakes and incomplete restructuring of sentences. Hopefully the manuscript is now at an acceptable level.